# Validated Finite Element Models of Premolars: A Scoping Review

**DOI:** 10.3390/ma13153280

**Published:** 2020-07-23

**Authors:** Raphaël Richert, Jean-Christophe Farges, Faleh Tamimi, Naim Naouar, Philippe Boisse, Maxime Ducret

**Affiliations:** 1Hospices Civils de Lyon, PAM Odontologie, 69007 Lyon, France; jean-christophe.farges@univ-lyon1.fr; 2Faculté d’Odontologie, Université de Lyon, Université Claude Bernard Lyon 1, 69100 Lyon, France; 3Laboratoire de Mécanique des Contacts et structures, UMR 5259 CNRS/INSA Lyon, 69100 Lyon, France; naim.naouar@insa-lyon.fr (N.N.); philippe.boisse@insa-lyon.fr (P.B.); 4Laboratoire de Biologie Tissulaire et Ingénierie thérapeutique, UMR 5305 CNRS/Université Claude Bernard Lyon 1, 69008 Lyon, France; 5Faculty of Dentistry, McGill University, QC H3A 2T5 Montreal, Canada; faleh.tamimimarino@mcgill.ca; 6College of Dental Medicine, Qatar Univerity, Doha 2713, Qatar

**Keywords:** systematic review, finite element analysis, premolar, operative dentistry, prosthodontics

## Abstract

Finite element (FE) models are widely used to investigate the biomechanics of reconstructed premolars. However, parameter identification is a complex step because experimental validation cannot always be conducted. The aim of this study was to collect the experimentally validated FE models of premolars, extract their parameters, and discuss trends. A systematic review was performed following Preferred Reporting Items for Systematic Reviews and Meta-Analyses (PRISMA) guidelines. Records were identified in three electronic databases (MEDLINE [PubMed], Scopus, The Cochrane Library) by two independent reviewers. Twenty-seven parameters dealing with failure criteria, model construction, material laws, boundary conditions, and model validation were extracted from the included articles. From 1306 records, 214 were selected for eligibility and entirely read. Among them, 19 studies were included. A heterogeneity was observed for several parameters associated with failure criteria and model construction. Elasticity, linearity, and isotropy were more often chosen for dental and periodontal tissues with a Young’s modulus mostly set at 18–18.6 GPa for dentine. Loading was mainly simulated by an axial force, and FE models were mostly validated by in vitro tests evaluating tooth strains, but different conditions about experiment type, sample size, and tooth status (intact or restored) were reported. In conclusion, material laws identified herein could be applied to future premolar FE models. However, further investigations such as sensitivity analysis are required for several parameters to clarify their indication.

## 1. Introduction

Endodontically treated premolars present one of the lowest survival rates, in particular owing to the high risk of vertical root fracture [1,2,3]. This fragility is mainly explained by the relatively small size of the premolar crown and the strong occlusal and lateral forces it is subjected to [1,4,5]. Clinical trials have investigated this topic, but they require a large number of patients to take into consideration the complexity and diversity of the clinical situations [6,7]. It thus appears necessary to find alternative ways of studying premolar behavior and gain sound scientific knowledge essential for elaborating effective tooth reconstruction protocols.

Finite element analysis (FEA), a computer-based method to solve engineering problems, has been commonly used to investigate mechanical performance in aeronautical and automotive fields, but also to evaluate biomechanical behavior in the medical domain, whether for prediction of osteoporotic fracture, temporomandibular replacement, or tooth reconstruction [8,9,10]. This numerical technique allows the development of patient-specific FEA, the measure of the impact of mechanical stress following force application, and the selection of the biomaterial most appropriate for a personalized clinical application [11,12]. Recent reviews have highlighted the increasing number of published papers reporting finite element (FE) models in oral medicine [10,13], especially for the analysis of new dental materials [12,13]. The development of a new FE model requires the definition of multiple parameters including, for example, the mesh, the material laws, and the boundary conditions [14,15,16]. Mesh parameters are used to describe how the dental volume is discretized using a specific number and type of element [15]. Material laws specify how the material will deform under masticatory forces [16]. Boundary conditions represent the loading of the tooth and the dental fixation [13,16]. Therefore, the use of validated parameters is required to avoid invalid conclusions [13,15,16] and the use of new materials too early in clinical practice. However, parameter identification is a complex step because no guidelines exist, and experimental validation cannot always be conducted [14].

The aim of the present paper was to collect the experimentally validated FE models of premolars and to extract their model parameters. A scoping review of the scientific literature was performed to summarize and discuss the usage trends of parameters that are most frequently used, to help create future models.

## 2. Materials and Methods

### 2.1. Protocol

The guidelines of the Preferred Reporting Items for Systematic Reviews and Meta-Analyses (PRISMA) statement [17] were followed to answer the study question: Which are the most frequently used parameters in experimentally validated FE models to simulate intact or restored premolars?

### 2.2. Information Sources and Search Strategy

Three electronic databases were searched (MEDLINE [PubMed], SciVerse Scopus, and The Cochrane Library) following the search strategy described in Table 1. The last search was performed on 16 October 2019, with no limit regarding the year of publication. Only records in the English language were considered. The records identified were imported from each database and saved into software (Excel Office 360, Microsoft, Redmond, WA, USA); duplicates were then removed using the corresponding software function.

### 2.3. Data Charting Process

Records were independently screened and evaluated for eligibility by two reviewers (R.R. and M.D.). Reasons for exclusion were noted in the software. Results of the two reviewers were then compared and discussed for final inclusion; in the case of conflict, a third person (P.B.) was consulted.

### 2.4. Screening

The titles and abstracts of records were screened for relevance to the study question. In order to provide homogeneity in the scoping review, records dealing with “Surgery or implantology analysis”, “Multiple prosthesis (splinted crowns, ribbon bonded, bridges, removable prosthesis),” “Orthodontic”, “Two-dimensional or axisymmetric models”, “Thermal analysis without mechanical load”, and “Studies that were not in English” or “Studies that did not present an abstract” were removed. In the case of inaccessible articles, authors were contacted by email and articles that were not accessible two months after request were excluded.

### 2.5. Eligibility

The different aspects of the study question were searched in the full text of the articles eligible for inclusion. The presence of a validation process with in vitro or in vivo tests was searched. Only studies that presented an error difference or comparison graphs between data obtained by FEA and experimental data were included. Those that were based on the evaluation of fracture areas or comparison to previously reported experimental data were excluded. In accordance with the PRISMA guidelines for scoping reviews, and owing to the lack of recommendations for FEA studies analysis, no quality assessment was performed [17].

### 2.6. Data Analysis

Twenty seven previously reported parameters were analyzed on included studies as follows: Study (objective, number of factors studied, presence of a statistical approach, failure criteria); construction of the model (technique to record the anatomy, presence of model for bone and ligament, number and type of elements, mesh quality assessment); material laws (enamel law, enamel Young’s modulus, enamel Poisson’s ratio, dentine law, dentine Young’s modulus, dentine Poisson’s ratio, bone law, ligament law); boundary and loading (type of loading, force intensity, force orientation); experimental comparison used for model validation (in vitro/in vivo condition, experimental test, comparison process, sample size, tooth type, loading head, tooth fixation)(16). The level of evidence of the included study was analyzed as previously reported [18].

## 3. Results

### 3.1. Selection of Sources of Evidence

Using the present search strategy, 1306 records were identified from the MEDLINE [PubMed], SciVerse Scopus, and The Cochrane Library databases (Figure 1). After removal of duplicates, 801 records remained for title and abstract screening. At this stage, records dealing with implantology (*n* = 231), multiple prosthesis (*n* = 143), orthodontics (*n* = 119), two-dimensional analysis (*n* = 40), thermal analysis (*n* = 17), or in a language other than English (*n* = 31) were excluded. Six records were subsequently excluded due to absence of the response to full text request, and 214 records were selected and read in full. Among these, 189 articles presented incomplete validation (no validation process was described in 136 articles, verification was done by the use of a convergence test only [*n* = 18] or mechanical tests without quantified comparison to numerical data [*n* = 35]). The remaining 25 (11.6%) articles presented a validated FE model by comparison to experimental data, and among these, those with two-dimensional validation of the FE models (*n* = 6) were excluded. Nineteen studies were finally included (Figure 1) and the main characteristics of the FE models were noted (Table 2) [19,20,21,22,23,24,25,26,27,28,29,30,31,32,33,34,35,36,37].

### 3.2. Characteristics of the Studies and Their Objective

All included studies were published over the past 20 years, with one team contributing to almost half of the included studies (*n* = 9, 47.4%). As all studies were performed in silico or were not randomized, these provided a low level of evidence (18). Regarding the objective followed in the included studies, direct coronal restorations were the most frequently analyzed (*n* = 8, 42.1%), followed by crown and post reconstructions (*n* = 5, 26.3%), intact tooth (*n* = 3, 15.7%), and restorations of cervical lesions (*n* = 3, 15.7%). Studies were focused on the influence of multiple therapeutic factors using the FE model (*n* = 16, 84.2%) or setting-up an FE model only (*n* = 3, 15.7%). The use of a statistical approach to evaluate the influence of parameters was reported in two studies (10.5%). The failure criteria was the principal stress (*n* = 13, 68.4%), the von Mises stress (*n* = 3, 15.7%), the strain tensor (*n* = 2, 10.5%), or the stress tensor (*n* = 1, 5.2%; Table 3).

### 3.3. Scoping Synthesis of Parameters

The FE models were designed using published data (*n* = 9, 47.4%), three-dimensional radiographic techniques (*n* = 7, 36.8%), and measurements on tooth slices (*n* = 3, 15.7%). A model of bone and ligament was present in 11 studies (57.9%), but no ligament or bone existed in five studies (26.3%), and ligament was simulated alone in three studies (15.7%). The number of elements in the mesh ranged from 840 elements for the oldest study to more than 500,000 elements in the three most recent studies. Eight studies (42.1%) presented a linear tetrahedral mesh, eight (42.1%) a hexahedral mesh, and three (15.7%) a quadratic tetrahedral mesh. The mesh quality was assessed in nine studies (47.4%) and only by the convergence test (Table 3).

All studies (*n* = 19) considered dentine and enamel to be homogeneous and linear elastic. Isotropic properties were used in 17 studies for enamel (89.5%) and in 16 studies (84.2%) for dentine, whereas orthotropic properties were used in other studies (enamel: *n* = 2; and dentine: *n* = 3). Regarding isotropic models (*n* = 17), Young’s modulus of the enamel was set at 84.1 GPa in eleven studies (64.7%), 41.4–48 GPa in three studies (17.6%), and 60.6–75 GPa in three studies (17.6%). Poisson’s ratio was set at 0.3–0.33 in 11 studies (64.7%) and 0.2–0.23 in six studies (35.3%). Regarding isotropic models for dentine (*n* = 16), Young’s modulus was set at 18–18.6 GPa in 15 studies (93.8%) and 15.4 GPa in one study (6.3%). Poisson’s ratio was set at 0.3–0.31 in 14 studies (84.5%) and 0.2–0.23 in two studies (12.5%). When cortical and cancellous bones were modeled (in 11 studies), isotropic and linear elastic conditions were used in 10 studies (90.9%) and orthotropic conditions were used in one study (9.1%). When the ligament was modeled (*n* = 14), it was considered isotropically linear elastic in most studies (*n* = 13, 92.9%) and nonlinearly visco-hyperelastic in one study (7.1%; Table 3).

The loading was most frequently simulated by a force applied on the top of the tooth in 18 studies (94.7%), whereas a contact with an indenter was modeled in one study (5.2%). When simulated, the applied force was axial in 15 studies (78.9%), oblique for two (10.5%), and in both directions in two studies (10.5%). The most frequently reported force intensity was 200 N (*n* = 7, 38.8%; Table 3).

The model validation was conducted in vitro on extracted teeth in the majority of studies (*n* = 18, 94.7%) except one, which was performed in vivo (5.2%). All predictions of the FE models were based on the comparison of tooth strains, but it was conducted using strain gauges (*n* = 13, 68.4%), a force sensor of a universal testing machine without strain gauges (*n* = 5, 26.3%), or an interferometer (*n* = 1, 5.2%). Difference was estimated mostly by calculating the mean squared error (*n* = 13, 68.4%) or by comparing experimental and numerical curves (*n* = 6, 31.6%). The sample size and tooth condition were not reported in one study (5.2%). When reported, the most frequent sample size was 5 (*n* = 8, 44.4%), and the most frequent condition was with a restoration (*n* = 9, 50.0%). The tooth fixation was not reported in two studies (10.5%). When reported, the most frequent tooth fixation was embedding in epoxy resin (*n* = 9, 52.9%). The loading head was not reported in nine studies (47.3%). When reported, the most frequent loading head was a 6 mm ball indenter (*n* = 7, 70%; Table 3).

## 4. Discussion

The present study identifies the experimentally validated studies. Almost all included studies reported similar parameters regarding dental material laws and validation based on in vitro evaluation of tooth strains. Nevertheless, other parameters dealing with the construction of FE models, boundary conditions, and experimental conditions revealed heterogeneity.

Despite the high number of screened articles, only a minority presented the chosen inclusion criterion of experimental validation. This is close to the 9% of validated FE models reported in a recently published review on dental implants [13], but much lower than in other biomedical fields where, for example, 39% of FE models on bone were experimentally validated [15]. This major issue should warn clinicians of their will to use recently developed materials reporting FEA, as results from non-validated simulations can be associated with inaccuracy and overinterpretation [13,15,38]. Furthermore, it is of note that in 24/189 articles, the authors stated that the model was validated, but this did not correspond to the definition used in aircraft certification [39,40] and in biomedical FEA [8], which is based on a quantified assessment between FE models and experiments. However, the results of this work should not be overinterpreted, because one team contributed to almost half of the included studies with the chosen inclusion criteria.

Almost all included studies investigated the influence of multiple clinical factors on stress, which confirms the complex biomechanical behavior of premolars [1,21,41]. However, a statistical approach to analyze the influence of each clinical factor was done in only two studies [21,25], although applied statistics have been reported to be useful to provide information on the sensitivity of an FE model to input factors and determine the presence of cofactors in the biomedical field [25,42,43]. Moreover, multiple failure criteria were reported to analyze stress in this review as observed in the literature [8,44,45]. To the best of our knowledge, no study has investigated which is the most adapted to the dental field. This is a major concern as it can introduce differences in fracture findings [8]. Furthermore, parameters related to the construction of the model reported heterogeneity, which could be associated with the evolution of technologies. The development of tomography has been sped up since the last decades, and this imaging procedure is now reported as being one of the most relevant techniques for recording accurate volumes in dentistry [9,13,15], thus defining a precise FE model [46]. Interestingly, the number of elements seems to have increased progressively in function of the rise in computing capacity, but it is of note that under half of the studies reported to have used a convergence test. Regarding the type of element, it is reported that as long as the mesh is sufficiently refined, either quadratic or linear tetrahedral elements could be used [8], but also that a lower number of quadratic tetrahedral elements was able to better simulate the stress distribution than linear tetrahedral ones [16,19]. However, the mesh quality was only assessed by a convergence test in the studies included herein, whereas other criteria could enable us to locally refine the mesh to avoid singularities and obtain a more continuous stress distribution [15]. This point is particularly important for the external surface of the root and the ligament where smoothing algorithms such as antialiasing were developed [47]. Furthermore, the bone and ligament were not always modeled herein as requiring complex laws [47,48,49,50], whereas their influence on the stress is now well reported [48].

Experimental studies have shown that dentine presents anisotropic properties [49]. However, almost all included studies use isotropic and linear elastic laws to define dentine, enamel, ligament, and bone. Isotropy and linearity appear well-adapted to simulate the premolar behavior on small deformations, but should be adjusted for other fatigue or crack propagation analyses [50,51]. This study helps to better understand and reproduce the mechanical behavior of dental structures, with the aim to develop materials that closely mimic their properties. Regarding enamel, values of Young’s modulus were herein heterogeneous, which raises an important question as enamel is reported to influence whole tooth deformation [36]. This result could be explained by the fact that FE models only consider predefined conditions with a set of fixed values [52], whereas uncertainties such as the difference in quality of enamel [49] or anatomical variations [52] exist between patients. This observation confirms that a consensus could not be defined for all parameters but that some parameters need to be adapted to create a patient-specific FEA [53]. Analysis of uncertainty and sensitivity is now required to determine the most appropriate values according to the clinical situation, as previously reported in other biomedical fields [14,15,54]. This mechanical question could meet the need of clinicians as patient-specific analysis was already reported to better report fractures than experienced clinicians [53]. Boundary conditions were also considered when comparing studies. Loading was mainly simulated by an axial force, but it mainly depends on if the clinician wants to evaluate the premolar and material’s behavior in compression or bending. This is a major concern as many studies have reported that the stress distribution [55,56] and the fracture strength [57] change considerably according to the occlusal loading for the premolar. A contact analysis was reported to enable a more patient-specific simulation by modeling the particular shape of the antagonist tooth [55], but this degree of complexity was considered only in one study [19].

Almost all included studies used in vitro tests to evaluate tooth strain, albeit being heterogeneous ways regarding experimental conditions. The strain gauge is a means that has been used for many years to evaluate tooth strains on a point [29], but interferometry enables the evaluation of the complete strain field at the tooth surface [36,58]. Interferometry implies more complex devices to obtain more information on the tooth deformations, but to our knowledge, no published paper exists to support whether such a complex method is more adapted than the strain gauge for model validation. Further investigations are required to define the experimental conditions adapted to each mechanical analysis. Parameters related to sample size, tooth type, tooth fixation, and loading were not always reported. There is a need to report all experimental conditions to facilitate study comparisons between research teams and the establishment of experimental guidelines. The 27-parameter list used herein was created according to reported considerations in biomechanics [16]. This list is related to the biomedical field in general and is non-exhaustive, but it could be adapted to more specific applications such as multi-scale or dynamic analysis by adding damping parameters for example [59].

## 5. Conclusions

The present study identifies the validated FE models for premolar analysis. Material laws identified herein seem to be an accepted trend and could be applied for future premolar FE models. Further investigations such as sensitivity analysis are required for several parameters to clarify their indications according to each patient.

## Figures and Tables

**Figure 1 materials-13-03280-f001:**
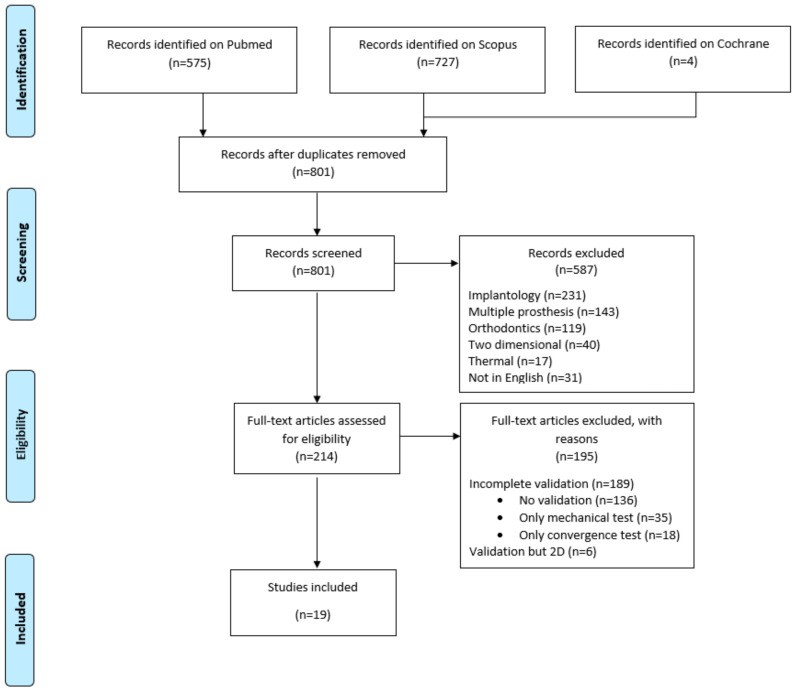
Flow diagram of the screening and selection process adapted from the Preferred Reporting Items for Systematic Reviews and Meta-Analyses (PRISMA) statement [17].

**Table 1 materials-13-03280-t001:** Electronic database and search strategy (16 October 2019).

Database	Search
MEDLINE [PubMed]	“finite element model premolar” OR “finite element analysis premolar” OR “finite element model premolar [Mesh]” OR “finite element analysis premolar [Mesh]”
SciVerse Scopus	“TITLE – ABS -KEY + finite + AND + element + AND + analysis + AND + premolar + OR + TITLE – ABS – KEY + finite + AND + element + AND + model + AND + premolar”
Cochrane Library	“TITLE – ABS -KEY + finite + AND + element + AND + analysis + AND + premolar + OR + TITLE – ABS – KEY + finite + AND + element + AND + model + AND + premolar”

**Table 2 materials-13-03280-t002:** Main characteristics and parameters of the included studies. CT refers to computed tomography, NURBS refers to Non-Uniform Rational Basis Splines, TET4 refers to a four-node tetrahedral element, TET10 refers to a ten-node tetrahedral element, HEX8 refers to an eight-node hexahedral element, GPa refers to gigapascal, mm refers to millimeter, N refers to newton, F/d refers to a measurement of force using a force sensor and displacement of the universal testing device, Exp/num refers to a quantified evaluation between experimental and numerical data, and σ/ε refers to stress/strain values.

Reference First Author Year	Aim	Technique	Number, Type of Elements, and Convergence	Law	Enamel	Dentine	Boundary	Loading	Experimental Comparison
[19] Limjeerajarus et al. 2019	Intact tooth, setting-up of a new FEM, principal stress	Micro CT, NURBS	1,062,233 TET10 Convergence	Elastic, linear	Orthotropic Ε_s_: 73.7 GPa ν: 0.23 Ε_c_: 63.3 GPa ν: 0.45?Ε_a_: 63.3 GPa ν: 0.23	Orthotropic Ε_s_: 17.1 GPa ν: 0.30?Ε_c_: 5.6 GPa ν: 0.33?Ε_a_: 5.6 GPa ν: 0.30	Ligament only	Axial, contact with a modeled indenter	F/d values in vitro Exp/num graphs 6.0 mm ball indenter, 30 intact teeth embedded in silicone
[20] MacHado et al. 2017	Cervical lesion, analysis of multi factors, von Mises stress	Scan, literature data, NURBS	1,709,931 TET10	Elastic, linear	Orthotropic Ε_s_: 73.7 GPa ν: 0.23 Ε_c_: 63.3 GPa ν: 0.45?Ε_a_: 63.3 GPa ν: 0.23	Orthotropic Ε_s_: 17.1 GPa ν: 0.30?Ε_c_: 5.6 GPa ν: 0.33?Ε_a_: 5.6 GPa ν: 0.30	Ligament only	Axial and oblique, forces: 150 N	σ/ε values in vitro Exp/num graphs 4.0 mm ball indenter, 25 intact teeth embedded in polyether
[30] Chang et al. 2015	Post and crown, analysis of multi factors, principal stress	Micro CT, segmentation	607,890 TET4 Convergence	Elastic, linear	Isotropic E: 84.1 GPa ν: 0.33	Isotropic E: 18.6 GPa ν: 0.31	Cortical spongy bone and ligament	Axial and oblique, forces: 200 N	σ/ε values in vitro Δε _exp/num_ < 6% 5 intact teeth embedded in epoxy resin
[31] Zelic et al. 2014	Coronal restorations, analysis of multi factors, principal stress	CT, segmentation	124,768 139,284 112,828 119,492 HEX8	Elastic, linear	Isotropic E: 84.1 GPa ν: 0.33	Isotropic E: 18.6 GPa ν: 0.31	Ligament only	Axial, force: 1025 N	F/d values in vitro Exp/num graphs 1 intact and 1 restored tooth embedded in silicone
[32] Guimarães et al. 2014	Cervical lesion, analysis of multi factors, principal stress	Measurement of tooth slices, NURBS	122,996 TET4	Elastic, linear	Isotropic E: 72.7 GPa ν: 0.33	Isotropic E: 18.6 GPa ν: 0.31	Cortical spongy bone and ligament	Axial, force: 105 N	F/d values in vitro Δε _exp/num_ < 4.6% 6.0 mm ball indenter, teeth embedded in epoxy resin
[33] Juloski et al. 2014	Post and crown, analysis of multi factors, principal stress	Scan, literature data, NURBS	31,240 TET4 Convergence	Elastic, linear	Isotropic E: 84.1 GPa ν: 0.33	Orthotropic Ε_s_: 25 GPa ν: 0.45 Ε_c_: 23.2 GPa ν: 0.29	Cortical spongy bone and ligament	Oblique, force: 200 N	σ/ε values in vivo Exp/num graphs one patient in vivo
[34] Lin et al. 2013	Post and crown, analysis of multi factors, principal stress	Micro CT, segmentation	134,810 HEX8 Convergence	Elastic, linear	Isotropic E: 84.1 GPa ν: 0.33	Isotropic E: 18.6 GPa ν: 0.31	Cortical spongy bone and ligament	Axial, force: 2000 N	σ/ε values in vitro Δε _exp/num_ = 18% 4 intact teeth
[35] Lin et al. 2009	Post and crown, analysis of multi factors, principal stress	Micro CT, segmentation	39,728 HEX8 Convergence	Elastic, linear	Isotropic E: 84.1 GPa ν: 0.33	Isotropic E: 18.6 GPa ν: 0.31	Cortical spongy bone and ligament	Axial, force: 100 N	σ/ε values in vitro Δε _exp/num_ < 10% 5 restored teeth embedded in epoxy resin
[36] Barak et al. 2009	Intact tooth, setting-up of a new FEM, strain alone	Micro CT, segmentation	438,638 TET4	Elastic, linear	Isotropic E: 75 GPa ν: 0.3	Isotropic E: 15 GPa ν: 0.3	No ligament or bone	Axial, force: 200 N	Interferometry Δε _exp/num_ = [11–85%] 4 intact teeth embedded in epoxy resin composite
[37] Lin et al. 2009	Post and crown, analysis of multi factors, principal stress	Micro CT, segmentation	39,728 HEX8 Convergence	Elastic, linear	Isotropic E: 84.1 GPa ν: 0.33	Isotropic E: 18.6 GPa ν: 0.31	Cortical spongy bone and ligament	Axial, force: 100 N	σ/ε values in vitro Δε _exp/num_ < 10% 5 restored teeth
[21] Lin et al. 2009	Coronal restorations, statistical analysis of multi factors, principal stress	Scan, literature data, NURBS	205,720 TET4	Elastic, linear	Isotropic E: 84.1 GPa ν: 0.2	Isotropic E: 18.6 GPa ν: 0.31	Cortical spongy bone and ligament	Axial and oblique, forces: 200 N	σ/ε values in vitro Δε _exp/num_ < 10% 5.0 mm ball indenter, 5 restored teeth embedded in epoxy resin
[22] Tajima et al. 2009	Intact tooth, setting-up of a new FEM, von Mises stress	CT, segmentation, NURBS	20,773 TET10	Elastic, linear	Isotropic E: 60.6 GPa ν: 0.3	Isotropic E: 18.3 GPa ν: 0.3	No ligament or bone	Axial, force: 88.3 N	σ/ε values in vitro Δε _exp/num_ = 6% 5 intact teeth embedded in dental stone
[23] Chang et al. 2008	Coronal restorations, analysis of multi factors, principal stress	Scan, literature data, NURBS	197,527 TET4 Convergence	Elastic, linear	Isotropic E: 84.1 GPa ν: 0.2	Isotropic E: 18.6 GPa ν: 0.31	Cortical spongy bone and ligament	Axial, force: 200 N	σ/ε values in vitro Δε _exp/num_ < 10% 6.0 mm ball indenter, 5 restored teeth embedded in epoxy resin
[24] Lin et al. 2008	Coronal restorations, analysis of multi factors, principal stress	Scan, literature data, NURBS	205,720 TET4	Elastic, linear	Isotropic E: 84.1 GPa ν: 0.33	Isotropic E: 18.6 GPa ν: 0.31	Cortical spongy bone and ligament	Axial, force: 200 N	σ/ε values in vitro Δε _exp/num_ < 10% 6.0 mm ball indenter, 5 restored teeth embedded in epoxy resin
[25] Lin et al. 2008	Coronal restorations, statistical analysis of multi factors, principal stress	Scan, literature data, NURBS	197,527 TET4 Convergence	Elastic, linear	Isotropic E: 84.1 GPa ν: 0.2	Isotropic E: 18.6 GPa ν: 0.31	Cortical spongy bone and ligament	Axial, force: 200 N	σ/ε values in vitro Δε _exp/num_ < 10% 6.0 mm ball indenter, 5 restored teeth embedded in epoxy resin
[26] Ausiello et al. 2004	Coronal restorations, analysis of multi factors, von Mises stress	Scan, literature data, NURBS	24,818 HEX8	Elastic, linear	Isotropic E: 48 GPa ν: 0.23	Isotropic E: 18 GPa ν: 0.2	No ligament or bone	Axial, force: 400 N	F/d values in vitro Exp/num graphs 6.0 mm ball indenter, 10 restored teeth embedded in composite
[27] Lee et al. 2002	Cervical lesion, analysis of multi factors, principal stress	Measurement of tooth slices, NURBS	5921 HEX8 Convergence	Elastic, linear	Isotropic E: 84.1 GPa ν: 0.2	Isotropic E: 18.6 GPa ν: 0.31	Spongy bone and ligament	Axial, force: 170 N	σ/ε values in vitro Δε _exp/num_ < 10% one intact tooth embedded in epoxy resin
[28] Ausiello et al. 2001	Coronal restorations, analysis of multi factors, von Mises stress	Scan, literature data, NURBS	7894 HEX8	Elastic, linear	Isotropic E: 48 GPa ν: 0.23	Isotropic E: 18 GPa ν: 0.2	No ligament or bone	Axial, force: 400 N	F/d values in vitro Exp/num graphs 6.0 mm ball indenter one restored tooth embedded in epoxy resin
[29] Toparli et al. 1999	Coronal restorations, analysis of multi factors, stress	Measurement of tooth slices, NURBS	840 HEX8	Elastic, linear	Isotropic E: 41.4 GPa ν: 0.3	Isotropic E: 18.6 GPa ν: 0.31	No ligament or bone	Axial, force: 300 N	σ/ε values in vitro Δε _exp/num_ < 10% 2.0 mm ball indenter, 2 restored teeth

**Table 3 materials-13-03280-t003:** Usage trends of parameters among validated models.

Section	Parameters	Most Frequently Used Choice	N/N_total—_%
Study	Objective	Coronal reconstruction	8/19–42.1%
	Number of factors studied	Multifactorial	16/19–84.2%
	Statistical approach	No statistical approach	17/19–89.5%
	Failure criteria	Principal Stress	13/19–68.4%
Model construction	Reconstruction technique	Literature data	9/19–47.4%
	Element type	TET4/HEX8	8/19–42.1%
	Mesh Quality	Convergence test	9/19–47.4%
	Presence of model for bone and ligament	Bone and ligament simulated	11/19–57.9%
Material law	Enamel law	Isotropy	17/19–89.5%
	Enamel Young’s modulus	84.1 GPa	11/17–64.7%
	Enamel Poisson’s ratio	0.3 or 0.33	11/17–64.7%
	Dentine law	Isotropy	16/19–84.2%
	Dentine Young’s modulus	18-18.6 GPa	15/16–93.8%
	Dentine Poisson’s ratio	0.3 or 0.31	14/16–84.5%
	Ligament law	Isotropy	10/11–92.9%
	Bone law	Isotropy	13/14–90.9%
Boundary and loading	Type of loading	Force	18/19–94.7%
	Force intensity	200 N	7/18–38.8%
	Force orientation	Axial	15/18–78.9%
Validation process	In vitro/in vivo	in vitro	18/19–94.7%
	Experimental test	Strain gauge	13/19–68.4%
	Comparison process	Exp/num error	13/19–68.4%
	Sample size	5 teeth	8/18–44.4%
	Tooth type	Restored	9/18–50.0%
	Tooth fixation	Epoxy resin	9/17–52.9%
	Loading	6 mm ball indenter	7/10–70.0%

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
