# Peer review of "Validated Finite Element Models of Premolars: A Scoping Review"

_materials, 2020, doi:10.3390/ma13153280_

Round 1
Reviewer 1 Report
Review Comments:
The authors have represented a review article entitled “Validated Finite Element Models for Premolar Reconstruction: A Scoping Review”. The aims and objectives of the manuscript is very good. The research is in early stage. This study needs some revision before accepted for publication.
Comments:
- Please try to explain the status of this review compared to presently reported literature.
- Please review Table 2. Among 19 reported references if we classified from country of origin, we can see 9 is from Taiwan. Please try to make more references with globally reported articles (try to include more countries), so that as a review paper we can get a global scenario.
- Please emphasize more clearly what is the importance of this review? How researchers will be benefitted?
- Please make a correlation between mechanical properties and the scope of this study with finite element model.
Author Response
Reviewer 1, Comment 1 - Please try to explain the status of this review compared to presently reported literature.
Authors: We would like to thank the reviewer for this comment. We have revised a sentence in the introduction to explain why this study is important for researchers. Indeed, there is no current guidelines for the use of FEA in the dental research field, and we propose in the paper a synthesis of the parameters that could be used with more confidence because they were mechanically validated.
Revision:
Development of a new FE model requires the definition of multiple parameters including, for example the mesh, material laws and boundary conditions [16].
…
However, the parameter identification is a complex step because no guidelines exist, and the experimental validation could not always be conducted [14].
Reviewer 1, Comment 2 - Please review Table 2. Among 19 reported references if we classified from country of origin, we can see 9 is from Taiwan. Please try to make more references with globally reported articles (try to include more countries), so that as a review paper we can get a global scenario.
Authors: PRISMA guidelines is a quality standard that aims to improve quality of review and we cannot add supplementary articles using this methodology. We also removed the country of origin in the table to prevent confusion regarding this information.
Revision:
However, the results of this work should not be overinterpreted because one team contributed to almost half of the included studies with the chosen inclusion criteria.
Reviewer 1, Comment 3 - Please emphasize more clearly what is the importance of this review? How researchers will be benefitted?
Authors: According to the comment, one sentence of the Introduction section and two sentences of the Discussion section were revised to explain why the results of this study could impact clinicians and researchers.
Revision:
Introduction section:
Recent reviews have highlighted the increasing number of published papers reporting finite element (FE) models in oral medicine [10,13], especially for the analysis of dental materials. Development of a new FE model requires the definition of multiple parameters including, for example the mesh, material laws and boundary conditions [16].
Discussion section:
only a minority reported a FE model … major issue should warn clinicians in their will to use recently developed material reporting FEA as results from non-validated simulations can be associated with inaccuracy and over interpretation.
This mechanical question could meet the need of clinicians as patient-specific analysis was already reported to better report fractures than experienced clinicians [53].
Reviewer 1, Comment 4 - Please make a correlation between mechanical properties and the scope of this study with finite element model.
Authors: A sentence has been added to the discussion to explain why the results of this study could impact the material properties.
Revision:
This study helps to better understand and reproduce mechanical behaviour of dental structures, with the aim to develop materials that closely mimic their properties.
Reviewer 2 Report
Even though the article seems to me well constructed and with solid bases. You correctly followed the prisma protocol, evaluated the evidence of the articles. However I miss some info about the role played by the external anatomy of the root and if FE models were studies to analyze it. In addition, the influence of this metod in reaserch and development of restorative materials should be enphatized.
Author Response
Reviewer 2, Comment 1 - Even though the article seems to me well constructed and with solid bases. You correctly followed the PRISMA protocol, evaluated the evidence of the articles. However I miss some info about the role played by the external anatomy of the root and if FE models were studies to analyze it.
Authors: We would like to thank the reviewer for this comment and have clarified this point by adding a sentence to the discussion to explain the impact of the mesh quality of the external surface of the root.
Revision:
This point is particularly important for the external surface of the root and the ligament where smoothing algorithms such as antialiasing were developed [47].
Reviewer 2, Comment 2 - In addition, the influence of this metod in reaserch and development of restorative materials should be emphasized.
Authors: We agree with this comment and have clarified this point by adding two sentences to the introduction and discussion to explain why this method play a major role in research and development of restorative materials.
Revision:
Introduction section:
Recent reviews have highlighted the increasing number of published papers reporting finite element (FE) models in oral medicine [10,13], especially for the analysis of dental materials. Development of a new FE model requires the definition of multiple parameters including, for example the mesh, material laws and boundary conditions [16].
Discussion section:
This study helps to better understand and reproduce mechanical behaviour of dental structures, with the aim to develop materials that closely mimic their properties.
Reviewer 3 Report
The paper is a scoping review which aims to find the most common problems with the study of premolar's fracture. The study of finite elements is a computerized technique that uses Mathematical schemes to find possible failures of a material or a component.
The fracture of premolar is a very common occurence, the study of it's weakness could be very useful for the clinician who needs to restore it's anatomy. It can be useful also to the technician who is going to build a prosthetic crown.
The authors made an appreciable work in search and preparing the paper which is linear and it can be helpful to researchers and clinicians.
However, it's written in a manner that a clinician would not understand since it's full of technical terms which can be very difficult to appreciate for who is not an engineer.
Since the paper can be useful to dentists and dental technician too, it can be appreciated if the authors could give detailed definition of the technical terms used like "linear tetrahedric mesh, convergence test, etc."
A more easy interpretation of the paper by clinicians could be useful also for other researcher who are interested in developing more resistant models of premolars due to avoid fracture.
Since this kind of investigations is usually conducted in mechanical engineering, the authors are kindly invited to clearly explain what could be difficult or impossible to understand to who is not an expert in finite element, like dentists and clinicians.
Author Response
Reviewer 3, Comment 1 - However, it's written in a manner that a clinician would not understand since it's full of technical terms which can be very difficult to appreciate for who is not an engineer. Since the paper can be useful to dentists and dental technician too, it can be appreciated if the authors could give detailed definition of the technical terms used like "linear tetrahedric mesh, convergence test, etc." A more easy interpretation of the paper by clinicians could be useful also for other researcher who are interested in developing more resistant models of premolars due to avoid fracture. Since this kind of investigations is usually conducted in mechanical engineering, the authors are kindly invited to clearly explain what could be difficult or impossible to understand to who is not an expert in finite element, like dentists and clinicians.
Authors: We would like to thank the reviewer for this comment. Accordingly, sentences have been added to the introduction, discussion, and conclusion to explain why the results of this study could impact clinicians and researchers. Additionally, several terms are now precociously defined to facilitate understanding of technical terms.
Revision:
Introduction section:
Recent reviews have highlighted the increasing number of published papers reporting finite element (FE) models in oral medicine [10,13], especially for the analysis of dental materials [12,13]. Development of a new FE model requires the definition of multiple parameters including, for example the mesh, material laws and boundary conditions [14,15,16]. Mesh parameters are used to describe how the volume is discretised using a specific number and type of element [15]. Material laws specify how the material will deform under masticatory forces [13,16]. Boundary conditions represent the loading and the dental fixation [13,16]. Therefore, the use of validated parameters is required to avoid invalid conclusions [13,15,16] and a too rapid use of new materials in clinical practice.
Discussion section:
This observation confirms that a consensus could not be defined for all parameters but that some parameters need to be adapted to create a patient specific FEA [53].
This mechanical question could meet the need of clinicians as patient-specific analysis was already reported to better report fractures than experienced clinicians [53].
Loading was mainly simulated by an axial force, but it mainly depends if the clinician wants to evaluate the premolar and material’s behaviour in compression or bending.
Conclusion section:
Further investigations such as sensitivity analysis are required for several parameters to clarify their indications according to each patient.
Reviewer 4 Report
The present study collects the experimentally validated FE models of premolars, extract their parameters and discuss the usage trends of parameters, which could help create models in the further study, utilizing the meta-analysis method.
- As author have shown, a heterogeneity was observed for several parameters associated to failure criteria and model construction. Simultaneously, the publication bias has not been evaluated. It is necessary that a sensitivity experiment should be added to increase the reliability of the study before publishment.
- There should be attached a list of abbreviation to the manuscript for readability. (for example, “PRISMA”, “F/d”, “Exp/num”)
- There are some spelling and syntax errors, especially in the results and discussion, which need correction. (for example. /)
Author Response
Reviewer 4, Comment 1 - As author have shown, a heterogeneity was observed for several parameters associated to failure criteria and model construction. Simultaneously, the publication bias has not been evaluated. It is necessary that a sensitivity experiment should be added to increase the reliability of the study before publishment.
Authors: We understand the point of the reviewer, but unfortunately the present study was a scoping review that aimed to collect parameters and describe the trend of this field. According to the recommendations for scoping review, bias evaluation is not mandatory because results of each study are not pooled and analysed [17].
Revision: No change
Reviewer 4, Comment 2 - There should be attached a list of abbreviation to the manuscript for readability. (for example, “PRISMA”, “F/d”, “Exp/num”)
Authors: According to this comment, we created a list of abbreviations and we added missing information regarding abbreviations in the text and figure legends.
Revision:
Main characteristics and parameters of the included studies. CT refer to computed tomography, NURBS refer to Non-Uniform Rational Basis Splines, TET4 refer to a four-node tetrahedral element, TET10 refer to a ten-node tetrahedral element, HEX8 refer to an eight-node hexahedral element, GPa refer to Gigapascal, mm refer to millimeter, N refer to newton, F/d refer to a measurement of force using a force sensor and displacement of the universal testing device, Exp/num to a quantified evaluation between experimental/ and numerical data and s/e to stress/strain.
Reviewer 4, Comment 3 - There are some spelling and syntax errors, especially in the results and discussion, which need correction. (for example. s/e)
Authors: The manuscript has been corrected following this comment.
Revision: Corrections were made throughout the text.
Round 2
Reviewer 3 Report
I think that the paper could be considered acceptable in present form.
Reviewer 4 Report
Accept in present form.